

**Atmospheric mixing ratios of methyl ethyl ketone (2-butanone) in tropical, bo-**
**real, temperate and marine environments**
**A. M. Yañez-Serrano[1,2], A. C. Nölscher[1,*], E. Bourtsoukidis[1], B. Derstroff[1], N. Zan-**
**noni[3], V. Gros[3], M. Lanza[4], J. Brito[5], S. M. Noe[6], E. House[7], C. N. Hewitt[7], B. Langford[8], E.**
**Nemitz[8], T. Behrendt[1,†], J. Williams[1], P. Artaxo[5], M. O. Andreae[1], and J. Kesselmeier[1]**
[1]{Biogeochemistry and Air Chemistry Departments, Max Planck Institute for Chemistry, P. O. Box 3060, D-55020
Mainz, Germany}
[2]{Instituto Nacional de Pesquisas da Amazônia (INPA), Av. André Araújo 2936, Manaus-AM, CEP 69083-000,
Brazil}
[3]{ Laboratoire des Sciences du Climat et de l'Environnement, LSCE/IPSL, CEA-CNRS-UVSQ, Université Paris-
Saclay, F-91191 Gif-sur-Yvette, France}
[4] {IONICON Analytik GmbH, Eduard-Bodem-Gasse 3, 6020, Innsbruck, Austria}
[5]{Instituto de Física, Universidade de São Paulo (USP), Rua do Matão, Travessa R, 187, CEP 05508-900, São
Paulo-SP, Brazil}
[6] {Institute of Agricultural and Environmental Sciences, Estonian University of Life Sciences, Kreutzwaldi 1, EE-
51014 Tartu, Estonia}
[7] {Lancaster Environment Centre, Lancaster University, Lancaster, LA1 4YQ, UK}
[8] {Centre for Ecology & Hydrology, Penicuik, EH26 0QB, UK}
* Now at Division of Geological and Planetary Sciences, California Institute of Technology, Pasadena, 91125 Cali-
fornia, USA.
† Now at Department of Biogeochemical Processes, Max Planck Institute for Biogeochemistry, Hans-Knöll-Str. 10,
D-07745, Jena, Germany.
Correspondence e-mail: a.yanezserrano@mpic.de. Phone: +55929984400709.
**Abstract**
Methyl ethyl ketone (MEK) enters the atmosphere following direct emission from vegeta-
tion and anthropogenic activities, as well as being produced by the gas-phase oxidation of volatile
organic compounds (VOCs) such as *n*-butane. This study presents the first overview of ambient
MEK measurements at six different locations, characteristic of forested, urban and marine envi-
ronments. In order to understand better the occurrence and behaviour of MEK in the atmosphere,
we analyse diel cycles of MEK mixing ratios, vertical profiles, ecosystem flux data, and HYSPLIT
back trajectories, and compare with co-measured VOCs. MEK measurements were primarily con-
ducted with proton transfer reaction – mass spectrometer (PTR-MS) instruments. Results from the
sites under biogenic influence demonstrate that vegetation is an important source of MEK. The diel
cycle of MEK follows that of ambient temperature and the forest structure plays an important role
in air mixing. At such sites a high correlation of MEK with acetone was observed (e.g. $r^2 = 0.96$
for the SMEAR-Estonia site in a remote hemi-boreal forest in Tartumaa, Estonia, and $r^2 = 0.89$ at



the ATTO pristine tropical rainforest site in central Amazonia). Under polluted conditions, we ob-
served strongly enhanced MEK mixing ratios. Overall, the MEK mixing ratios and flux data pre-
sented here indicate that both biogenic and anthropogenic sources contribute to its occurrence in
the global atmosphere.

**Keywords:** Methyl ethyl ketone, plant emission, anthropogenic sources, air quality, rainforest, bo-
real forests.

**1.  Introduction**
Methyl ethyl ketone ($C_4H_8O$; MEK, also known as 2-butanone, butanone, methyl acetone,
butan-2-one, methylpropanone, ethylmethylketone and methylacetone) is an oxygenated volatile
organic compound (OVOC). Its occurrence in the atmosphere has been reported over a wide range
of environments (Cecinato et al., 2002; Hellén et al., 2004; Ho et al., 2002; Kesselmeier et al.,
1997; Kim et al., 2015; McKinney et al., 2011; Singh et al., 2004) with typical mixing ratios of
0.03 - 4 ppb (Ciccioli and Mannozzi, 2007; Kim et al., 2015). Although often being measured
alongside other volatile organic compounds (VOCs), atmospheric MEK has received little attention
to date. The photochemistry of acetone may serve as an example of how ketones affect the compo-
sition and chemistry of the atmosphere by delivering free radicals to the upper troposphere (Colomb
et al., 2006; Finlayson-Pitts and Pitts, 2000; McKeen et al., 1997) and thus increasing the ozone
formation potential and altering the oxides of nitrogen ($NO_x$) regime (Ciccioli and Mannozzi, 2007;
Folkins et al., 1998; Prather and Jacob, 1997). This understanding may be transferred to MEK, as
this molecule is structurally similar to acetone with a comparable absorption spectrum (Martinez
et al., 1992). Several studies report that the mixing ratio of MEK in the free troposphere is roughly
one quarter of that of acetone (Moore et al., 2012; Singh et al., 2004). However, MEK is about an
order of magnitude more reactive than acetone with respect to the hydroxyl radical (Atkinson,
2000), which makes it a compound of interest in ongoing discussions about the inability to fully
account for the reactivity of OH (Nölscher et al., 2016).
There are several known but poorly characterized sources of MEK to the atmosphere. Ter-
restrial vegetation (Bracho-Nunez et al., 2013; Brilli et al., 2014; Davison et al., 2008; De Gouw





et al., 1999; Isidorov et al., 1985; Jardine et al., 2010; Kirstine et al., 1998; König et al., 1995;
McKinney et al., 2011; Ruuskanen et al., 2011; Song and Ryu, 2013; Steeghs et al., 2004; Wilkins,
1996; Yáñez-Serrano et al., 2015), fungi (Wheatley et al., 1997) and bacteria (Song and Ryu, 2013;
Wilkins, 1996) are known to emit MEK. It is also emitted directly by several anthropogenic
sources, including biomass burning (Andreae and Merlet, 2001), solvent evaporation (Le Calvé, et
al., 1998; Kim et al., 2015; Legreid et al., 2007) and vehicle exhaust (Bon et al., 2011; Brito et al.,
2015; Guha et al., 2015; Liu et al., 2015; Verschueren, 1983). In addition, MEK can be formed via
the atmospheric oxidation of other compounds (de Gouw et al., 2003; Jenkin et al., 1997; Neier
and Strehlke, 2002; Sommariva et al., 2011).
Looking in more detail at biogenic sources, MEK emissions have been observed from dif-
ferent types of vegetation, including forest canopies (Brilli et al., 2014; Jordan et al., 2009; Yáñez-
Serrano et al., 2015), pasture (Davison et al., 2008; De Gouw et al., 1999; Kirstine et al., 1998) and
clover (De Gouw et al., 1999; Kirstine et al., 1998). The MEK production and release mechanisms
are manifold, but poorly understood. Studies show higher MEK emissions after cutting and drying
of leaves than under no-stress conditions (Davison et al., 2008; De Gouw et al., 1999). Due to the
water solubility of MEK in leaves and on surfaces (Sander, 2015), Jardine et al. (2010) suggested
MEK emissions to be dependent on evaporation from storage pools in leaves. It has been suggested
that MEK takes part in tri-trophic signalling following herbivore attack (Jardine et al., 2010; Song
and Ryu, 2013). The roots of plants have also been found to release MEK in root-aphid interactions
(Steeghs et al., 2004). Decaying plant tissue may also act as a source of MEK to the atmosphere
(Warneke et al., 1999). Furthermore, some studies indicate the importance of MEK emissions by
microbes, such as *Brevibacterum linens, Bacillus spp* and thermophilic gram-positive actinomy-
cetes bacteria (Song and Ryu, 2013; Wilkins, 1996), and fungi such as *Trichoderma spp* (Wheatley
et al., 1997).
MEK does not only enter the atmosphere via direct emissions, but also results from the
atmospheric photooxidation of VOCs such as *n*-butane, 2-butanol, cis-2-butene/pentene, 3-methyl
pentane and 2-methyl-1-butene (de Gouw et al., 2003; Jenkin et al., 1997; Neier and Strehlke, 2002;
Sommariva et al., 2011). Although butane in the atmosphere comes predominantly from anthropo-
genic sources (Kesselmeier and Staudt, 1999), some studies have reported emission of *n*-butane
from vegetation (Donoso et al., 1996; Greenberg and Zimmerman, 1984; Hellén et al., 2006; König



et al., 1995; Zimmerman et al., 1988). The MEK yield from *n*-butane oxidation is ~ 80% (Singh et
al., 2004). It is important to note that no mechanistic pathways have been found for atmospheric
MEK production from the dominant biogenic VOCs isoprene, α- and β-pinene and methyl butenol
oxidation (Rollins et al., 2009; Singh et al., 2004).
In the atmosphere MEK reacts mainly with OH ($k_{OH}=1.15 \times 10^{-12}\,cm^3\,s^{-1}$) (Chew and Atkin-
son, 1996), while reactions with $O_3$ and $NO_3$ are very slow during the day and hence negligible
(Atkinson and Arey, 2003). MEK has a lifetime of 5.4 days at an OH concentration of $1.6 \times 10^6$
radicals $cm^{-3}$, whereas isoprene and acetone have lifetimes of 8.2 h and 38 days, respectively, under
the same conditions (Grant et al., 2008). The atmospheric degradation of MEK leads to acetalde-
hyde and formaldehyde formation. In the presence of $NO_x$, MEK can lead to peroxyacetyl nitrate
(PAN) and ozone formation (Grosjean et al., 2002; Pinho et al., 2005). In the upper troposphere,
MEK photolyzes and regenerates OH (Atkinson, 2000; Baeza Romero et al., 2005; De Gouw et
al., 1999) as does acetone, potentially increasing ozone formation.
Anthropogenic biomass burning leads to significant MEK emissions, of about 2 Tg $a^{-1}$ glob-
ally (Andreae and Merlet, 2001 and unpublished updates; Schauer et al., 2001). Furthermore, about
9 Tg $a^{-1}$ of other C4 compounds are emitted by biomass burning, which may act as MEK precursors.
Another strong source of MEK is biofuel and charcoal combustion, with emissions of ~830 mg $kg^{-1}$
$^1$ of dry biomass (compared to an emission rate of ~260 mg $kg^{-1}$ of dry mass for biomass burning
of savannah and grassland vegetation types, Andreae and Merlet, 2001). Despite the fact that bio-
mass burning emission rates have been fairly well characterized, vehicular emissions, food cook-
ing, industrial activities, cigarette smoke and other anthropogenic sources have not been character-
ized. Even though MEK is present in urban atmospheres, there are no observations of MEK emis-
sions from vehicles. MEK is also emitted by chemical plants as it is widely used in industry as a
solvent, and is toxic (Le Calvé, et al., 1998), but not carcinogenic (National Center for Biotechnol-
ogy, 2015).
Here we report recent findings on MEK from six different sites, including biogenic and
anthropogenic dominated environments, in order to understand MEK sources in different environ-
ments. Our large dataset allows a closer view of this important, almost ubiquitous species in Earth's
atmosphere.



## 2. Sites and Methodology


The field sites compared in our study cover areas from pristine to remote anthropogenically
influenced tropical forests, as well as boreal and Mediterranean regions. Measurements were per-
formed by proton transfer reaction – mass spectrometry (PTR-MS) and partly complemented by
gas chromatography - flame ionization detector (GC-FID) and gas chromatography - mass spec-
trometry (GC-MS) analytical techniques (Figure 1, Table 1).
Online MEK measurements were performed with quadrupole PTR-MSs (Ionicon Analytic
GmbH, Austria, Lindinger et al., 1998) at all sites, except for CYPHEX where a PTR-Time-Of-
Flight-MS (PTR-ToF-MS, Ionicon Analytic GmbH, Austria, Lindinger et al., 1998) was used. The
PTR-MSs were operated at standard conditions (2.2 mbar drift pressure, 600 V drift voltage, 142
Td for ATTO and SMEAR-Estonia; 2.0 mbar drift pressure, 550 V drift voltage, 129 Td, for TT34;
2.2 mbar drift pressure, 600 V drift voltage, 135 Td for $O_3HP$; 2.2 mbar drift pressure, 560 V drift
voltage, 132 Td for T2 and 2.2 mbar drift pressure, 600 V drift voltage, 137 Td for CYPHEX).
Periodic background measurements and weekly humid calibrations were performed at all
sites. Gravimetrically prepared multicomponent standard were obtained from Apel & Riemer,
USA, for ATTO, TT34, T2 and CYPHEX, and from Ionicon Analytik GmbH, Austria, for $O_3HP$
and SMEAR-Estonia.

### 2.1. The Amazon Tall Tower Observatory, ATTO: pristine tropical rainforest (Amazon, Brazil).


The Amazon Tall Tower Observatory (ATTO) site is located in central Amazonia, 150 km
NE of Manaus, Brazil (Figure 1) within a pristine primary tropical rainforest. The site is equipped
with a tall tower (325 m) and two 80 m towers. One of them (02°08'38.8" S, 58°59'59.5" W) is a
80-m walk-up tower, where the trace gas measurements take place. It is surrounded by a forest with
a canopy height of approximately 35 m and with at least 417 different tree species among 7293
screened trees of $\geq$ 10 cm diameter at breast height (DBH) in the twelve 1-ha inventoried plots
(Andreae et al., 2015). The climate of this site is typical for tropical rainforests with a drier season
(July-October) and a wet season (December-April, Nobre et al., 2009).





Measurements for this study took place 18 February - 15 March 2014. They were carried

out at seven different heights (0.05, 0.5, 4, 24, 53 and 79 m) for 2 minutes at each height. The inlet
lines were made of PTFE, 9.5 mm OD, insulated and heated to 50 ºC and had PTFE particle inlet
filters. More information about the gradient system and PTR-MS operation at ATTO can be found
elsewhere (Nölscher et al., 2016; Yáñez-Serrano et al., 2015).

Additionally, ambient samples for off-line measurements with GC-FID were taken on 11

March 2014 from 08:30 to 11:00 LT. They were collected at 24 m using a GSA SG-10-2 personal
sampler pump and adsorber tubes (Carbograph 1, Carbograph V 130 mg of Carbograph 1 (90 $m^2$
$g^{-1}$) followed by 130 mg of Carbograph 5 (560 $m^2$ $g^{-1}$)). The size of the Carbograph particles was
in the range of 20–40 mesh. Carbograph 1 and 5 were provided by Lara s.r.l. (Rome, Italy). Samples
were collected for 20 min with a flow of 167 ml $min^{-1}$ passing about 3.3 l of ambient air through
the adsorbent. Cartridges were transported to the laboratory for analysis by a Perkin Elmer Auto-
system XL GC-FID. These samples generally matched the results of the PTR-MS. For details on
sampling see Kesselmeier et al., (2002).
**2.2. TT34: remote tropical rainforest (Amazon, Brazil)**

The ZF2 site is located in the Reserva Biologica do Cuieiras in central Amazonia, 60 km

NNW of Manaus (2°35'39.4"S 60°12'33.4"W) within a remote primary tropical rainforest (Figure
1). The site is equipped with two towers, TT34 and the K34. The TT34 triangular tower is 40 m
high and embedded within the forest with a canopy height of approximately 30 m. The biodiversity
of this site is also high and the climate is very similar to that at the ATTO site. More information
about the site can be found elsewhere (Karl et al., 2009; Martin et al., 2010).

Measurements for this study were made from 1 September 2013 to 20 July 2014 at 41 m,

at a fast rate (0.5 Hz) for virtual disjunct eddy covariance (vDEC) flux derivations techniques (Karl
et al., 2002; Langford et al., 2009; Rinne et al., 2002). Wind vector data were obtained with a sonic
anemometer (Gill R3, USA) mounted at the top of the tower close to the PTR-MS inlet. The PTR-
MS inlet line was made of PFA (12.7 mm OD) (PFA-T8-062-100, Swagelok), and was insulated
and heated to 40 ºC inside the air-conditioned cabin.
**2.3. Station for Measuring Ecosystem-Atmosphere Relations, SMEAR-Estonia: re-**
**mote hemi-boreal forest (Tartumaa, Estonia)**



The Station for Measuring Ecosystem-Atmosphere Relations (SMEAR-Estonia) site is lo-
cated in the Järvselja Experimental forest station in Tartumaa, SE Estonia (58°16'N 27°16'E),
within a remote hemi-boreal zone, far from major anthropogenic disturbances (Noe et al., 2011,
Figure 1). The site is equipped with a tower of 24 m height. The canopy height is about 16-20 m
and the remote hemi-boreal forest consists of a mixture of tree species, with Norway spruce *(Picea*
*abies)* dominating. The climate is boreal with a growing season of 170-180 days. More information
about the site can be found elsewhere (e.g. Bourtsoukidis et al., 2014a; Noe et al., 2011, 2016).
The measurements were made between 3 and 17 October 2012. Sampling was done using
a dynamic, automated glass enclosure with measurement cycles of 36 seconds. The inlet line (9.5
mm) was made of glass and was insulated and heated to 70ºC. A dynamic exchange enclosure was
used to measure emission rates from a Norway spruce branch located in the upper canopy at 16 m.
While the focus of this study was the quantification of emission rates from a Norway spruce tree,
ambient mixing ratios were derived as well using the box model described in Bourtsoukidis et al.,
(2014b).
Furthermore, at SMEAR-Estonia, off-line measurements with a GC-MS were carried out
for periods of three days each in June and July 2012, with samples taken every 4 hours at two
heights (2 m and 20 m). Samples for GC-MS analysis were also taken from cuvettes enclosing
some common plant species at the site (Table 1). In addition, VOC emissions from soil litter were
monitored monthly. The air samples were drawn into multi-bed stainless steel cartridges (10.5 cm
length, 3 mm inner diameter, Supelco, Bellefonte, PA, USA) filled with Carbotrap C 20/40 mesh
(0.2 g), Carbopack C 40/60 mesh (0.1 g) and Carbotrap X 20-40 mesh (0.1 g) adsorbents (Supelco).
Even though the site usually experiences low ozone mixing ratios of 10 - 30 ppb (Noe et al., 2012),
a catalytic Cu(II) ozone scrubbing system (Sun et al., 2012) was applied. Three constant-flow air
sample pumps (1003-SKC, SKC Inc., Huston, TX, USA) and one multisample constant-flow air
sample pump (224-PCXR8, SKC Inc., Huston, TX, USA) allowed four samples to be collected at
the same time. Each sample took 30 min with a flow of 200 ml min$^{-1}$ concentrating 6 l of ambient
air onto the adsorbent. More information can be found elsewhere (Noe et al., 2012).
**2.4. Observatoire de Haute Provence, O$_3$HP: rural Mediterranean temperate forest**
**(Provence, France)**



The oak observatory (O₃HP, https://o3hp.obs-hp.fr) at the "Observatoire de Haute Pro-
vence" (OHP, http://www.obs-hp.fr/welcome.shtml), is located within a rural Mediterranean tem-
perate forest in the French Mediterranean region, 60 km north of Marseille, the closest large city
(43°55'54.0" N 5°42'43.9" E, Figure 1). A 10 m mast was set up inside the oak forest with a canopy
height of approximately 5 m. The O₃HP site is dominated by *Quercus pubescens Willd* (75% of
trees) and *Acer monspessulanum L.* (25%) forming a sparse canopy, while *Cotinus coggygria Scop.*
and other grass species constitute the understorey canopy. The climate at the site is typical Medi-
terranean, with dry and hot summers and humid and cool winters. More information about the site
can be found elsewhere (Genard-Zielinski et al., 2015; Kalogridis et al., 2014).
The measurements took place during 29 May - 12 June 2014 as part of the CANOPEE
project (Biosphere-atmosphere exchange of organic compounds: impact of intra-canopy pro-
cesses). Ambient measurements were carried out at 2 m (inside the canopy) on consecutive days in
intervals of 5 minutes. The 9.5 mm Teflon inlet lines were insulated and heated above ambient
temperature and had no particle filter. In addition, light non-methane hydrocarbons (from ethane
to hexane) were measured with a GC-FID (Chromatotec, Saint-Antoine, France) in-line with the
PTR-MS as described in Zannoni et al. (2016).
**2.5. T2: mixed urban and rainforest influenced environment (Amazon, Brazil)**
The T2 site is part of a set of experimental sites within the GoAmazon project to study the
effect of the pollution plume from the city of Manaus on the otherwise pristine Amazonian atmos-
phere. The T2 site is located 8 km downwind, i.e. to the west, of Manaus (3°8'21.12" S, 60°7'53.52"
W, Figure 1). Given its location, near Manaus and across the Rio Negro, air mass transport to the
sampling site is strongly modulated by a river breeze, alternating between mostly biogenic condi-
tions, resulting from the surrounding forest, and the city emissions. The climate is tropical and
similar to that at the ATTO and ZF2 sites.
The measurements for this study took place between 15 February and 15 November 2014.
They were carried out at 12 m above the laboratory container with 30 minute cycles. The inlet line
was made of insulated Teflon (9.5 mm OD) without PFTE particle filter.
**2.6. CYPHEX: mixed marine, rural environment influenced by aged air masses (Cy-**
**prus)**





The Cyprus Photochemistry Experiment (CYPHEX) campaign took place at a site located
in the NW inshore part of Cyprus, in the Paphos region (34º57'50.0" N, 32º22'37.0" E) (Figure 1).
The site experiences mixed marine and rural emissions influence. The climate is Mediterranean,
warm and dry, and shrubs and small trees dominate the sparse vegetation.
The measurements took place during July and August 2014 without a single rain event.
Instruments were installed inside containers and connected to a stack inlet that reached up 5 m
above the container roofs. Air was drawn through the 15 m stack inlet of 0.5 m with high flow rate
(5 l min$^{-1}$) to minimize wall losses. The subsampling inlet line was made of Teflon (13 mm OD),
was insulated and heated to 35 ºC, and had a PTFE inlet particle filter.
**3.  Results**
**3.1. Sites dominated by biogenic emissions**
All the pristine or remote sites studied were characterized by relatively low mixing ratios
of nitrogen oxides (NO$_x$) (< 3 ppb of nitrogen dioxide (NO$_2$) for O$_3$HP (Kalogridis et al., 2014;
Zannoni et al., 2016), 0.2-0.8 ppb of NO$_x$ for SMEAR-Estonia, and <1 ppb NO$_x$ for the Amazon
rainforest (Kuhn et al., 2010)). The diel cycles of MEK at these sites followed a comparable pattern
(Figure 2), where MEK mixing ratios were highest in the middle of the day, following the maxima
of light and air temperature. The dominant source at these sites was considered to be biogenic.
Mixing ratios of MEK correlated well with ambient temperature (r$^2$=0.57 (ATTO), r$^2$=0.83
(SMEAR), r$^2$=0.47 (O$_3$HP)), while it was less well correlated with photosynthetically active radi-
ation (PAR) (r$^2$= 0.23 (ATTO), r$^2$=0.26 (SMEAR), r$^2$=0.67 (O$_3$HP)). This suggests that ambient
temperature predominantly influenced MEK emission rates from plants.
The vertical observations at ATTO revealed a strong diel variability of the magnitude and
vertical distribution of MEK mixing ratios throughout the forest canopy and in the atmosphere
above. Figure 3 shows an example of one day (7$^{th}$ of March 2014) hourly vertical profile of MEK
from 13:00-15:00 LT, from the ground to the atmosphere, suggesting that the canopy top is the
major source of MEK at the site on such days. Such figures were found for 83% (for the afternoon
hours) and 45% (for the morning hours) of all days of measurements. In addition, MEK mixing
ratios decreased significantly beneath the canopy towards the forest floor, possibly due to dry dep-
osition or generally smaller emissions due to less light and temperature.





At the TT34 rainforest site, ecosystem-scale fluxes were directly calculated from the PTR-
MS measurements using the method of virtual disjunct eddy covariance (vDEC) (Karl et al., 2001b;
Figure 4). The fluxes averaged over the entire 11-month measurement period (covering both parts
of dry and wet season) clearly demonstrate an emission of MEK by the rainforest during daytime
with the highest emissions around noon, and no emissions during nighttime.
Online ambient mixing ratios of MEK, as measured by the PTR-MS in the hemi-boreal
forest at the SMEAR-Estonia site during autumn 2012, were on average $0.15 \pm 0.04$ ppb (range
0.09 - 0.25 ppb). These mixing ratios are almost a factor of 2 lower than ATTO and $O_3HP$ during
daytime hours. This difference among boreal forests and broad-leafed forests (of ATTO and $O_3HP$)
could be related to the temperature dependence of MEK emissions apparently common among all
biogenic sites.
The rural Mediterranean temperate forest site at $O_3HP$ differs significantly from the tropical
rainforest (ATTO, TT34) or the hemi-boreal forest (SMEAR-Estonia, Figure 1). The trees at $O_3HP$
are predominantly *Quercus pubescens*, a high isoprene emitter (Keenan et al., 2009). At this site,
the exchange of air through the forest canopy is enhanced because the canopy is sparse. As shown
in Figure 2, ambient MEK mixing ratios measured inside the canopy (2 m) increased with temper-
ature in the morning. During the day, increased forest emissions of MEK seemed to balance the
rise of the boundary layer depth, resulting in a plateau until sunset. The fluctuation of MEK after
sunset may be understood as a result of a ceased source revealing the deposition as it can hardly be
explained by gas-phase chemistry or the reduced nocturnal boundary layer height.
During the CANOPEE campaign at the $O_3HP$ site, additional GC-FID samples were taken
at 2 m, allowing measurements of several anthropogenic light hydrocarbons, including n-butane.
This sampling was performed in parallel to the PTR-MS measurements. All samples contained *n*-
butane, which was of anthropogenic origin. The MEK signal of the PTR-MS did not show any
covariance with the *n*-butane signal as measured by the GC-FID. Hence, MEK at the $O_3HP$ site
could not be related to the atmospheric oxidation of *n*-butane. Furthermore, the absence of a cor-
relation with other anthropogenic tracers let us conclude that MEK at this site was predominantly
of biogenic origin.



The measurements obtained by PTR-MS at the presented sites dominated by biogenic emis-
sions were occasionally confirmed by GC-FID and GC-MS, which are compound selective. At
ATTO the same range of MEK mixing ratios for the same hour of the day and height for the GC-
FID and the PTR-MS measurements was found, indicating that the PTR-MS signal was only or at
least dominated by MEK. To identify sources, canopy measurements at SMEAR-Estonia were
complemented by emission measurements using cuvettes with GC-MS identification. Common
hemi-boreal forest species, such as *Quercus robur, Tilia cordata, Sorbus aucuparia, Betula pu-*
*bescens* and *Picea abies*, were screened for VOC emissions. The highest emissions of MEK were
found from *Tilia cordata* and *Picea abies* (Table 2). The data match those reported by
(Bourtsoukidis et al., 2014a) who measured an emission rate of MEK of $2.6\pm2.2$ ng $g_{(dw\ needle)}^{-1}$ $s^{-1}$
from *Picea abies* using PTR-MS. The GC-MS technique obtained a very similar value of 2.3 ng
$g_{(dw\ needle)}^{-1}$ $s^{-1}$. MEK emissions from *Picea abies* were relatively small compared to other VOCs
emitted by the same plant species such as total monoterpenes and acetone which reached 93.2 and
27.6 ng $g_{(dw\ needle)}^{-1}$ $s^{-1}$, respectively (Bourtsoukidis et al., 2014a). In addition to plant sources, the
emissions of MEK from leaf litter were quantified with GC-MS. MEK litter emissions, with a
monthly average of 7 µg $m^{-2}$ $h^{-1}$, were of comparable magnitude to the emissions of MEK from the
screened hemi-boreal forest tree species, including *Quercus robur* or *Sorbus aucuparia*, which
emitted 8 - 9 µg $m^{-2}$ $h^{-1}$ of MEK (Table 2).

### 3.2. Anthropogenically influenced sites

Anthropogenically influenced sites are characterized by air masses that have passed over
polluted cities or industrially used regions. This air typically has elevated mixing ratios of $NO_x$,
other products of fossil fuel combustion such as aromatic compounds, carbon monoxide (CO), and
higher aerosol loading. Regional biomass burning plumes can also be a source of anthropogenic
input to air masses and are here considered as anthropogenic.
The T2 data set was sorted for polluted periods (air masses loaded with CO, black carbon,
high aerosol loading, aromatic compounds) and non-polluted periods. Periods with CO higher than
130 ppb during the tropical wet season and higher than 160 ppb during the dry season were con-
sidered polluted. As shown in Figure 5, MEK mixing ratios strongly increase with pollution. The
T2 site in Brazil is located on the bank of the Rio Negro and is affected by both, the tropical rain-



forest (biogenic) and the megacity of Manaus (anthropogenic). The location of the T2 site down-
wind of Manaus and upwind of the rainforest minimizes the biogenic influence. MEK mixing ratios
were generally lower for the clean conditions at T2 than mixing ratios found at ATTO or TT34
(Figure 2 and 5). Nevertheless, the mixing ratios of MEK during polluted conditions ($0.7 \pm 0.33$
ppb during dry season at 13:00 LT) reached or even exceeded those at the other tropical rainforest
sites ($0.32 \pm 0.13$ ppb at 13:00 LT for ATTO and $0.45 \pm 0.28$ ppb at TT34; Figure 2 and 5). Pre-
sumably, anthropogenically affected air as transported across the Rio Negro from the city of Ma-
naus (~2 million inhabitants; IBGE, 2014) generated a plume with a mixture of anthropogenic
MEK directly emitted in Manaus or MEK formed by oxidation of e.g. *n*-butane during transport.
The general trend observed in Figure 6 is an increase of MEK mixing ratios when easterly winds
came from Manaus (located to the East of T2). However, there were times when winds blew from
the North, and incident air masses passed through large rainforest areas, in which TT34 is included,
these air masses then crossed the river and arrived at the site). During these times, when air masses
were mostly dominated by biogenic emissions, MEK enhancement reached on average 200 ppt.

Mixing ratios of MEK at T2 were found to be significantly enhanced during polluted con-

ditions for both dry and wet season (Figure 7). The relative enhancement within polluted periods
at 13:00 LT ranged around a factor of 1.5 for the wet season and of 1.8 for the dry season. During
the dry season, the day-to-day variability was more intense, as reflected by the standard deviations
which increased by 360 % for the clean conditions and 410 % for the polluted conditions relative
to the wet season clean and polluted values, respectively. This may indicate a difference in the
sources and sinks regulating MEK mixing ratios among the different seasons. Examples of this
difference could be an increase of MEK due to biomass burning, more abundant during the dry
season, or changes in the deposition rates due to changes in rain frequency.

The CYPHEX campaign took place at Ineia, North-West Cyprus, at a location that has very

little significant vegetation nearby. The air masses that pass through the site are either from Western
Europe passing across France and Spain and then the Mediterranean Sea, or Eastern Europe (e.g.
Turkey, Greece). During the CYPHEX campaign, the hourly median MEK mixing ratios did not
show any distinct diel cycle or relations to temperature or net radiation (Figure 5) strongly suggest-
ing no significant local sources were present. Furthermore, backward air mass trajectories, as cal-
culated by the HYSPLIT model (NASA, USA) (Figure 8), can be used to delineate times when



Cyprus was affected by easterly and westerly flow. On average, easterly air masses contained 0.13
± 0.03 ppb whereas westerly masses contained 0.08 ± 0.02 ppb. This difference can be due to
differences in source strength, the greater duration of boundary layer transport from the west and
hence marine uptake, or to photochemical loss during transport.

### 3.3. Compilation of measurement data

In order to investigate the origin and characteristics of MEK in the atmosphere, we calcu-
lated the determination coefficient ($r^2$) between the mixing ratios of MEK and other co-measured
VOC species for each site (Table 3). This coefficient is the ratio of the variability of the MEK
mixing ratios over the variability of the other VOC mixing ratios available for each site. Acetone,
acetaldehyde, monoterpenes, isoprene, isoprene oxidation products and methanol (Kesselmeier and
Staudt, 1999; Laothawornkitkul et al., 2009) are regarded as being of biogenic origin. Compounds
such as benzene, toluene, xylene and acetonitrile are considered as typical anthropogenic tracers
(Andreae and Merlet, 2001; Finlayson-Pitts et al., 1997).
The determination coefficients ($r^2$) between MEK and other VOCs may indicate similarities
of production and consumption pathways. In general, biogenic sites, namely ATTO, SMEAR-Es-
tonia, and $O_3$HP, showed relatively high correlations between MEK and almost all biogenic VOCs
($r^2 > 0.5$). Exceptions are the $r^2$ of isoprene, monoterpene and isoprene oxidation products for $O_3$HP.
For instance, the highest determination coefficient was found for MEK and acetone at the
SMEAR site ($r^2 = 0.97$). In SMEAR-Estonia overall high correlations were found with the oxygen-
ated compounds, acetone, acetaldehyde and methanol, as well as with monoterpenes and isoprene.
At ATTO, correlations were only slightly lower. The determination coefficients calculated for the
$O_3$HP observations were generally lower than for ATTO and SMEAR-Estonia, further influenced
by the higher turbulent mixing due to sparser vegetation, leading to a quick oxidation. Nevertheless,
the good correlations of MEK with typical biogenically emitted compounds, such as isoprene, iso-
prene oxidation products, monoterpenes, methanol and acetone, corroborated the biogenic origin
of MEK emissions at the biogenic sites.
At the anthropogenically influenced sites, T2 and CYPHEX, determination coefficients for
the biogenic compounds were generally lower, apart from the $r^2$ (0.64 and 0.45, respectively) be-
tween MEK and acetone. It is important to note that although T2 is a mixed anthropogenic and



biogenic site, the determination coefficient for MEK and acetone was high, but very low for the
rest of the biogenic compounds. For the anthropogenic compounds, T2 had an $r^2$ of 0.27 for MEK
and acetonitrile and MEK and xylene. Furthermore, the data from the Cyprus site showed poor
correlation of MEK with any biogenic compound, but a correlation coefficient of $r^2 = 0.58$ for
MEK and toluene, an anthropogenic tracer.
**4.  Discussion**
**4.1. PTR-MS measurements**
Most of the measurements in this study were performed with a quadrupole PTR-MS, a
technique that monitors selected VOC ions, online and with fast time response. A disadvantage is
the separation by masses with a mass resolution of only 1 amu. For some masses, several com-
pounds and/or compound-fragments may be detected as one signal. The quadrupole PTR-MS sig-
nal at *m/z* 73 is attributed to MEK, but may have contributing signals of water clusters (de Gouw
et al., 2007), butanal (Inomata et al., 2010; McKinney et al., 2011; Slowik et al., 2010; Warneke et
al., 2007), acrylic acid (de Gouw et al., 2003), 2-methyl propanal (Baraldi et al., 1999; Jardine et
al., 2010), and methyl glyoxal (Holzinger et al., 2007; Jordan et al., 2009). We have tried to take
into account possible interferences by using different analytical techniques and supplementary in-
formation. At the SMEAR-Estonia site, the accompanying GC-MS observations validated the sig-
nal for MEK. Additionally, the GC-FID samples taken at ATTO corroborated the signal for MEK
at this site. Nevertheless, we try to give a short overview below about the interferences of other
trace gases with the PTR-MS identification of MEK.
Methyl glyoxal is a likely contributor to the observed signal at the PTR-MS protonated
mass *m/z* 73, especially in areas where there are high levels of isoprene. It is formed following the
oxidation of methyl vinyl ketone and methacrolein, which are both isoprene oxidation products
(Calvert and Madronich, 1987; Lee et al., 2006). Supported by GC-FID measurements and rela-
tively low isoprene levels during the wet season (Yáñez-Serrano et al., 2015), we can assume that
the contribution of methyl glyoxal to this mass was insignificant at the rainforest sites (ATTO and
TT34). Furthermore, at O3HP the correlation between MEK and the isoprene oxidation products
was low ($r^2 = 0.41$). Despite the high isoprene emissions it seems that these oxidation products and



methyl glyoxal did not significantly contribute to the signal at *m/z* 73. During the CYPHEX cam-
paign the PTR-ToF-MS could unambiguously distinguish between MEK and methyl glyoxal (at
73.0648 amu and 73.0284 amu respectively).

Even though a contribution of butanal to *m/z* 73 of up to 65% (Lindinger et al., 1998) and

20% (Williams et al., 2001) has been reported previously, most butanal fragments on *m/z* 57 (Ion-
icon Analytic GmbH). Acrylic acid, a marine compound (Liu et al., 2016) that may interfere at *m/z*
73, was probably not of relevance at sites under biogenic influence. However, in the case of an-
thropogenically influenced sites, such as T2, interferences may have been of relevance. Karl et al.
(2007) and Ciccioli et al., (2014) measured tropical biomass burning emissions and found that *m/z*
73 is comprised of 74% MEK and 23% 2-methyl propanal (73.1057 amu). Even though none of
the sites presented in this study was severely influenced by biomass burning, we cannot completely
rule out a possible direct emission of 2-methyl propanal by plants, which is of lower magnitude
than from biomass burning (Hafner et al., 2013; Jardine et al., 2010; Karl et al., 2005). Due to the
standard operation conditions of the PTR-MS under our measurement conditions, we neglected
water clusters as they are regarded to be insignificant (McKinney et al., 2011; Yáñez-Serrano et
al., 2015). Summarizing these issues, we note that several studies have concluded *m/z* 73 to origi-
nate from MEK only (Bourtsoukidis et al., 2014a; Crutzen et al., 2000; De Gouw et al., 1999, 2000;
Holzinger et al., 2000; Karl et al., 2001a, 2005; Kim et al., 2015; Millet et al., 2015; Steeghs et al.,
2004). Based on these considerations and the similarity of magnitudes measured by the PTR-MS
as compared with the GC results, we assume *m/z* 73 is representative of the atmospheric MEK
present.

### 4.2. The biogenic MEK

The data obtained at the biologically influenced sites demonstrated that MEK was emitted

by vegetation. This is clearly supported by the canopy-scale net flux observations of MEK at the
TT34 rainforest site (Figure 4) as well as the diel cycles of the mixing ratios at the other biogeni-
cally influenced sites (Figure 2). Furthermore, the cuvette-level measurements at SMEAR-Estonia
also corroborated the MEK emission by vegetation. In addition, a contribution by other biogenic
sources such as dead and decaying plant matter was also observed at SMEAR-Estonia to be of
similar magnitude to boreal plant species emissions, and indicating a source from plant litter, in
accordance with the results from Warneke et al., (1999) that measured MEK emission from the



abiotic processes of plant decaying matter. This is not the case for the tropical sites where vertical
profiles show canopy emissions dominate.
High correlation coefficients suggested strong relations between the emission processes for
MEK and other biogenic compounds (Table 3). A similar approach has been used previously by
Goldstein and Schade (2000) to unveil the sources of acetone. Similarly, Davison et al. (2008)
found a high correlation coefficient between MEK and acetone of $r^2=0.87$ and a relatively poor
correlation between MEK and monoterpenes ($r^2=0.54$). They surmised that good correlations indi-
cated a common origin for biogenically emitted compounds. Furthermore, a resemblance of the
pattern of acetone and MEK has been reported for the ATTO site before (Yáñez-Serrano et al.,
2015). In our study, we found high determination coefficients for MEK with acetone and MEK
with temperature, and lower $r^2$ for MEK and compounds such as isoprene and monoterpenes for
all the biogenic sites (Table 3). This could indicate that MEK forest emissions are more related to
processes resembling acetone emissions and temperature dependent processes, rather than light and
temperature dependent emission mechanisms, as for isoprene and monoterpenes (Jardine et al.,
2015; Kesselmeier and Staudt, 1999).
Plant physiological production pathways have been reported for MEK formation. MEK can
be formed, similarly to acetone, as a by-product of a cyanohydrin lyase reaction during cyanogen-
esis (Fall, 2003; Vetter, 2000). This chemical defence pathway was also identified in clover by
Kirstine et al. (1998) and de Gouw et al. (1999) as a result of mechanical stress, and can be of
special importance for tropical rainforests (Miller et al., 2006). On the other hand, in places such
as SMEAR-Estonia, dominating plant species are not cyanogenic, and other processes for MEK
formation are probably more dominant. In pine trees, acetone is produced from light-dependent
and –independent processes that can be associated with the decarboxylation of acetoacetate occur-
ring in microorganisms and animals (Fall, 2003), oxidation of fatty acids leading to ketone emis-
sions (Niinemets et al., 2014), from pyruvic acid leading to acetyl-CoA (Kesselmeier and Staudt,
1999), or produced from uncharacterized biochemical reactions (Fall, 2003). Such processes could
also be related to MEK emissions.
Even though extensive laboratory measurements are needed to identify the dominant plant
process or processes responsible for MEK emission, this study demonstrated the role that temper-
ature can exert on such emissions. Hence, forests around the world may act as very different sources



for atmospheric MEK. This can be seen for boreal forests (SMEAR-Estonia), with distinctly lower
temperatures, where MEK levels were significantly lower. However, other factors must be consid-
ered, such as Leaf Area Index (LAI) and plant species composition, as well as the environmental
factors, water availability and mechanical stress. Mechanical stress has already been observed by
de Gouw et al. (1999) to act as a driver for MEK emissions. This is in close agreement with in-
creased emissions of MEK as observed at the SMEAR site in Estonia during the installation of the
branch enclosure, causing a disturbance of the branch during the installation of the dynamic cham-
ber (Bourtsoukidis et al., 2014a).
Due to its relatively long atmospheric lifetime (~5 days for the reaction with OH; Grant et
al., 2008), MEK is expected to accumulate in the atmosphere until removal. Hence, atmospheric
mixing ratios can reflect seasonality and changes in dominating sources, affected by radiation,
temperature and phenology, from more biogenic dominance during the wet season to transport
phenomena and oxidation processes of primarily emitted compounds from regional biomass burn-
ing, as it has been seen in 2013 at the ATTO site (Yáñez-Serrano et al., 2015). Additionally, the
canopy structure seems to be important for air mixing and fast oxidation, as seen for the $O_3HP$ site
with an apparently faster mixing due to sparser vegetation and consequent dampening of the am-
plitude of the diel cycle. Furthermore, due to its oxygenated nature, partitioning to and from aque-
ous surfaces is likely, including deposition and surface reactions. Its high water solubility might
allow dissolution within leaf water (Sander, 2015) triggering bidirectional exchange of MEK
(McKinney et al., 2011; Niinemets et al., 2014). Due to its high solubility in water and its relatively
long lifetime, MEK could potentially influence gas-aqueous reactions on aerosol surfaces (Nozière,
2005). This has been shown indirectly by the production of methyl glyoxal after its oxidation by
OH, having implications for the formation of organics in the aerosol aqueous phase (Rodigast et
al., 2015).

### 4.3. The anthropogenic MEK

A clear difference could be observed between the anthropogenic and biogenic influenced
sites presented in this study. The T2 site represented a site with mixed influence by urban area and
tropical rainforest. Affected by anthropogenic and biogenic sources ambient mixing ratios of MEK
were higher than at the pristine ATTO rainforest site. Polluted episodes (from the Manaus plume)
with an increase of MEK could be distinguished for both, the wet and the dry season, suggesting a



short range transport of air masses. On the other hand, when the wind is blowing from the North,
MEK mixing ratios were also present, showing an influence from biogenic forest emissions (Figure
7), thus having a mix of biogenic and anthropogenic influence at the T2 site. A strong seasonality
of MEK mixing ratios at T2 reflected biomass burning as a common occurrence in the Amazon
region during the dry season (Artaxo et al., 2013). In addition to MEK, a higher contribution of
butanal affecting $m/z$ 73 (Inomata et al., 2010; Karl et al., 2007) might be possible, although MEK
has been reported to have a much higher emission factor (range from 0.17 to 0.83) than butanal
(range from 0.04 to 0.21) for biomass burning (Andreae and Merlet, 2001).

We regarded CYPHEX as an anthropogenically influenced site with weak or no apparent

direct sources, but affected by anthropogenic air masses after long range transport over marine
areas. Losses by transport over the sea and chemical decomposition led to the lowest MEK mixing
ratios of all compared sites. Determination coefficients of MEK with the biogenic tracers were
relatively poor for T2 and CYPHEX. However, determination coefficients were also poor for the
anthropogenic tracers, although higher than these coefficients at the biogenic sites. MEK showed
highest correlation with acetone, indicating similar sources and fate in air mixing and chemistry
processes. MEK transported over a long distances (10 days) is lost by photochemical aging or
deposition as evidenced by the lowest values reported from CYPHEX. This is despite known sec-
ondary photochemical sources, i.e. n-butane oxidation (Katzenstein et al., 2003; Kwok et al., 1996).
Interestingly, even under polluted conditions, MEK did not correlate with aromatic compounds,
except during CYPHEX, although this correlation deteriorated in the more aged westerly air
masses. This can only be understood as a result of a very complex mixture of anthropogenic sources
of MEK which vary from direct emission by industry (Legreid et al., 2007), gasoline combustion
(Verschueren, 1983), biomass burning (Andreae and Merlet, 2001), night-time anthropogenic ac-
tivities (Guha et al., 2015) and vehicular emissions (Brito et al., 2015). Furthermore, chemical
processing during transport may contribute, such as oxidation of $n$-butane, however, the longer
transport times during CYPHEX from the west corresponded to lower values.
**5. Remarks and conclusions**

The comparison of MEK mixing ratios in different parts of the world is necessary in order

to understand how this ubiquitous compound occurs and behaves in the atmosphere. MEK can lead





to PAN and ozone formation in the atmosphere (Pinho et al., 2005) and photochemical odd-hydro-
gen production in the upper troposphere (Atkinson, 2000; Baeza Romero et al., 2005; De Gouw et
al., 1999) which can further enhance the MEK ozone forming potential (Folkins et al., 1998;
Prather and Jacob, 1997). Furthermore, as higher mixing ratios of MEK have been found under
polluted conditions, human exposure to this toxic compound should be considered(Le Calvé, et al.,
1998). Of the widely used atmospheric chemistry models, only GEOS-Chem explicitly computes
MEK but only with regard to anthropogenic origin. On the basis of the data presented here from
forest sites, it is necessary for atmospheric chemistry models to also include biogenic MEK emis-
sions to better estimate its effects on the environment. Sites under biogenic influence showed
marked diel variability, matching biogenic VOC emissions and temperature. Structural forest fea-
tures seem to affect turbulent mixing and diluting of trace gases like MEK, as in the case of $O_3HP$
with patchy vegetation. MEK seemed to be produced in plants in a similar fashion to acetone, likely
released during mechanical stress. Possible pathways for productions in plants are oxidation of
fatty acids, cyanogenesis, production from pyruvic acid leading to Acetyl-CoA, light-dependent
and –independent processes that can be associated with the decarboxylation of acetoacetate occur-
ring in microorganisms and animals.
This study presents the first compilation and comparison of ambient measurements of MEK
at different sites. MEK patterns and mixing ratios differ around the globe depending on sources
and transport. Vegetation and litter have been identified as sources of MEK and magnitude of
sources varied among the tropical rainforest, the Mediterranean temperate forest and the hemi bo-
real forest following a likely temperature dependence. However, via different filtering methodolo-
gies (CO filtering and backward trajectories), the anthropogenic input from polluted regions, such
as the mixed urban & tropical rainforest and mixed marine environmentis, is often found to be the
dominant contribution.

## 6. Acknowledgements

For ATTO: We thank the Max Planck Society and the Instituto Nacional de Pesquisas da Amazonia for con-
tinuous support. Furthermore, we acknowledge the support by the ATTO project (German Federal Ministry of Educa-
tion and Research, BMBF funds 01LB1001A; Brazilian Ministério da Ciência, Tecnologia e Inovação FINEP/MCTI
contract 01.11.01248.00); UEA and FAPEAM, LBA/INPA and SDS/CEUC/RDS-Uatumã. We would like to thank
especially all the people involved in the logistical support of the ATTO project, in particular Reiner Ditz and Hermes
Braga Xavier. We acknowledge the micrometeorological group of INPA/LBA for their collaboration concerning the



meteorological parameters, with special thanks to Marta Sá, Antonio Huxley and Leonardo Oliveira. We would like
to acknowledge Stefan Wolff for the construction, support and maintenance of the inlet system. We are grateful to
Nina Knothe for logistical help. We would also like to thank Thomas Klüpfel for all the great support provided with
the PTR-MS operation in the laboratory as well as in the field. This paper contains results of research conducted under
the Technical/Scientific Cooperation Agreement between the National Institute for Amazonian Research, the State
University of Amazonas, and the Max-Planck-Gesellschaft e.V.; the opinions expressed are the entire responsibility of
the authors and not of the participating institutions.
For TT34: We thank the Natural Environment Research Council for funding the CLAIRE-UK project (refer-
ence NE/I012567/1), A. Valach, B. Davison and M. Shaw for assistance and E. Nemitz, B. Langford and A.R. Mac-
Kenzie for valuable discussions.
For SMEAR: We would like to acknowledge the EU Regional Development Foundation: "Environmental
Conservation and Environmental Technology R&D Programme" project BioAtmos (3.2.0802.11-0043), "Internation-
alization of Science Programme" project INSMEARIN (10.1-6/13/1028), and the "Estonian Research Infrastructures
Roadmap" project Estonian Environmental Observatory (3.2.0304.11-0395). We express our gratitude towards the
Archimedes foundation (international program DoRa) and the "Freunde und Förderer der Goethe Universität" that
funded E.B. for conducting research in Estonia. We would like to additionally thank Dominika Radacki, Javier Roales,
Beate Noe, Eero Talts, Ahto Kangur and Miguel P. Estrada for providing valuable help with the setup and transporta-
tion. Special thanks to Boris Bonn for the insightful discussions and comments during the production of this article.
For O$_3$HP: The measurements presented in this study were supported by the European Commission's 7th
Framework Programme under Grant Agreement Number 287382 "PIMMS", ANR-CANOPEE and ChArMEx, CEA
and CNRS. We acknowledge B. Bonsang and C. Kalogridis for the GC-FID measurements, J.P. Orts and I. Reiter for
logistical support J. Lathière for managing the CANOPEE project.
For T2: We thank Bruno Takeshi for all the logistical support. Furthermore, we acknowledge the support by
FAPESP grant 2013/25058-1 e 2013/05014-0.

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




Table 1: Measurement sites, site environment, sampling dates, methods used and sampling heights.

| Site | Characteristics | Canopy height | Time of Sampling | Technique | Type of measurement | Measurement heights |
|---|---|---|---|---|---|---|
| ATTO (Brazil) | Pristine tropical rainforest | 35 m dense veg. | Feb/Mar 2014 | PTR-MS | Ambient | 0.05, 0.5, 4, 24, 38, 53, 79 m |
| | | | | GC-FID (samples for off-line analysis, collected volume=3.34 L) | | 24 m |
| TT34 (Brazil) | Remote tropical rainforest | 30 m dense veg. | Sep 2013 – Jul 2014 | PTR-MS | Ambient, including canopy-scale fluxes | 41 m |
| SMEAR (Estonia) | Remote hemi-boreal forest | 16-20 m dense veg. | Jun, Jul, Oct 2012 | GC-MS (samples for off-line analysis, collected volume=6 L) | Ambient, plant, soil enclosure | 2, 20 m |
| | | | Oct 2012 | PTR-MS | Ambient, plant enclosure | 16 m |
| O₃HP (France) | Rural temperate forest | 5 m sparse veg. | May-Jun 2014 | PTR-MS | Ambient | 2 m |
| T2 (Brazil) | Mixed urban and rainforest influenced environment | Influence from veg. nearby | Feb-April 2014 July-Oct 2014 | PTR-MS | Ambient | 14 m |
| CYPHEX (Cyprus) | Mixed marine, rural environment influenced by aged air masses | None, on top of a hill | Jul-Aug 2014 | PTR-TOF-MS | Ambient | 8 m |




Table 2: Emission rates of MEK for typical hemi-boreal plant species and soil litter measured by GC-MS technique at the SMEAR site.

| Plant species and soil cuvettes | Mean $\mu g\ m^{-2}\ h^{-1}$ | Standard deviation $\mu g\ m^{-2}\ h^{-1}$ | Standard error $\mu g\ m^{-2}\ h^{-1}$ | Median $\mu g\ m^{-2}\ h^{-1}$ | Number of data points for statistics |
|---|---|---|---|---|---|
| *Quercus robur* | 8.12 | - | - | - | 1 |
| *Tilia cordata* | 12.93 | 4.89 | 3.46 | 12.93 | 3 |
| *Sorbus aucuparia* | 9.08 | - | - | - | 1 |
| *Betula pubsecens* | 9.36 | 5.10 | 2.94 | 8.21 | 3 |
| *Picea abies* | 13.76 | 5.05 | 2.91 | 15.51 | 3 |
| Leaf litter | 7.00 | 3.37 | 2.11 | 6.58 | 24 |

Table 3: Determination coefficient ($r^2$) of MEK and other co-measured VOC at the sites investigated. Green indicates sites with biogenic influence and red sites with anthropogenic influence. In the second column, the mean noon mixing ratios are expressed in ppb. The correlations above 0.5 are colour coded with warmest colours for highest determination coefficients. Determination coefficients for the TT34 site in Amazonia are missing due to lack of data.

| | $r^2$ | Acetone | Acetaldehyde | Monoterpenes | Isoprene oxidation products | Methanol | Isoprene | Acetonitrile | Benzene | Toluene | Xylene |
|---|---|---|---|---|---|---|---|---|---|---|---|
| | | | | Biogenic tracers | | | | Anthropogenic tracers | | | |
| **Biogenic sites** | **TT34** | - | - | - | - | - | - | - | - | - | - |
| | **SMEAR** | 0.97 | 0.89 | 0.72 | - | 0.90 | 0.84 | - | - | - | - |
| | **ATTO** | 0.89 | 0.62 | 0.75 | 0.75 | 0.51 | 0.77 | 0.49 | 0.07 | 0.27 | 0.04 |
| | **O₃PH** | 0.61 | 0.62 | 0.12 | 0.41 | 0.57 | 0.14 | 0.19 | 0.03 | 0.15 | 0.00 |
| **Anthropogenic sites** | **T2** | 0.64 | 0.21 | - | 0.41 | 0.27 | 0.06 | 0.27 | 0.11 | 0.07 | 0.27 |
| | **CYPHEX** | 0.45 | 0.42 | 0.07 | 0.10 | 0.25 | 0.08 | 0.00 | 0.58 | 0.09 | - |





## 9. Figures

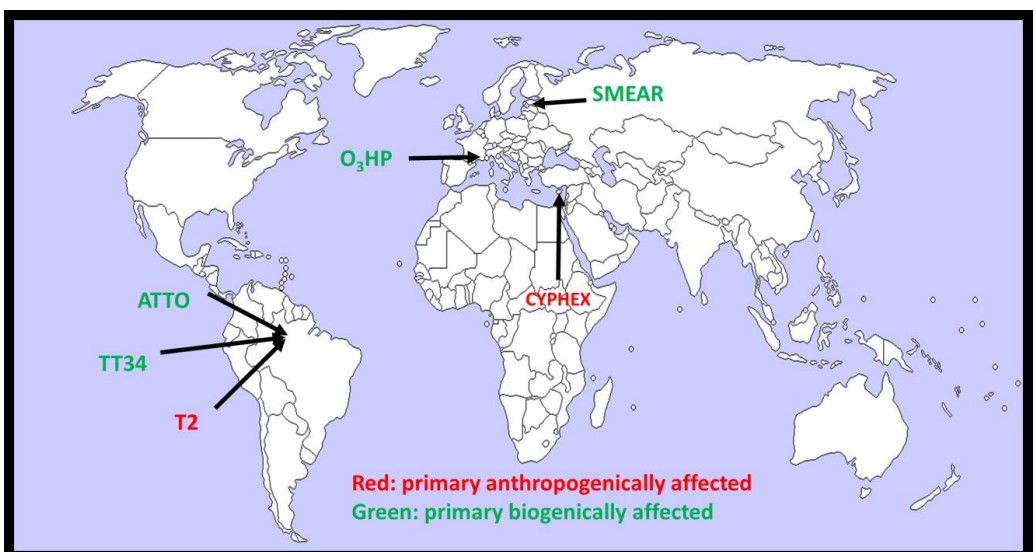

Figure 1: World map showing the location of the different sites. The names are colour coded depending on if they have primarily biogenic influence (green) or a primarily anthropogenic influence (red). Wiki-media Foundation, 2016.

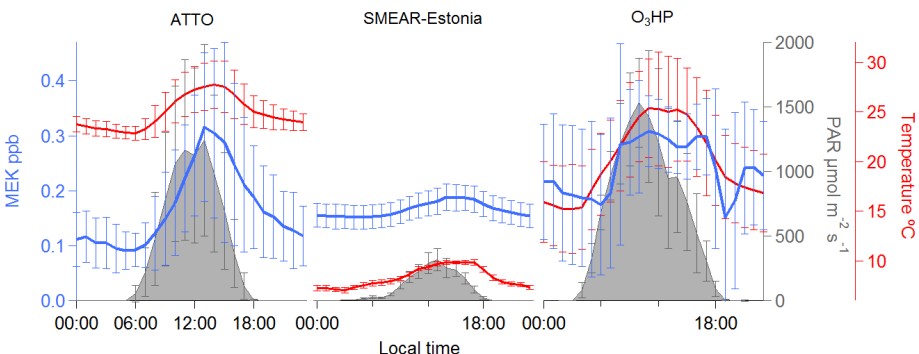

Figure 2: Hourly average diel cycles of MEK at the ATTO (left), SMEAR-Estonia (middle) and O$_3$HP (right) sites, for the period of measurements (wet season 2014 for ATTO at 38 m, May and June 2014 for O$_3$HP at 2 m, and October 2014 for SMEAR-Estonia at 16 m). Hourly mean diel cycles of temperature and PAR are also shown in red and grey, respectively. Error bars represent the standard deviations.





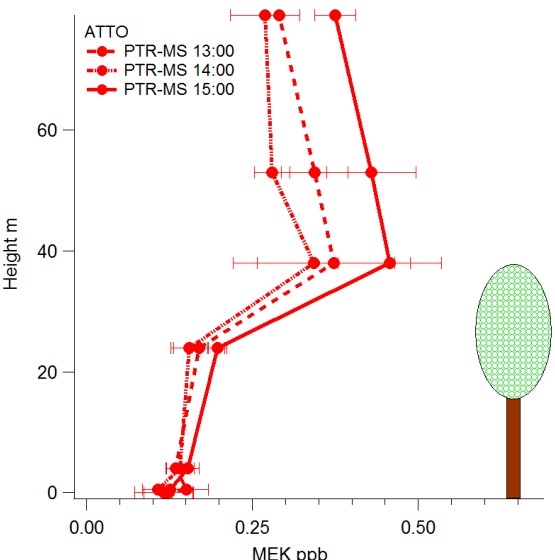

Figure 3: Hourly average vertical profiles of MEK mixing ratios at ATTO for the 7th of March 2014 for 13:00 LT (dashed lines), 14:00 LT (dotted and dashed lines) and 15:00 LT (thick lines). Error bars of vertical profiles are the standard deviations.

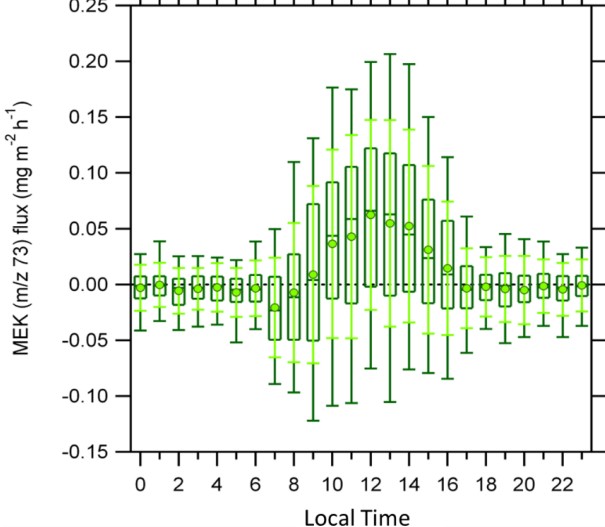

Figure 4: Hourly average MEK fluxes at the TT34 tower for the period Sept 2013 - July 2014. The light green circles represent means and associated error bars are one standard deviation. The central line of the box plots (dark green) indicates the median, bottom and top lines the 25th and 75th percentile respectively and whiskers are the 5th and 95th percentiles.




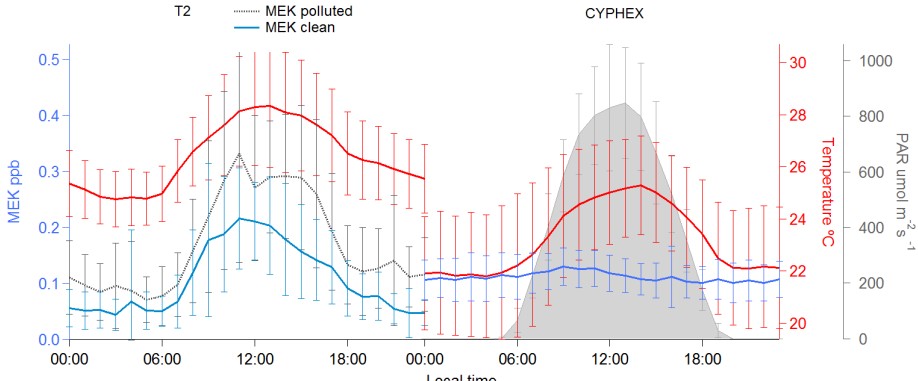

Figure 5: Hourly average diel cycles of MEK at the T2 (left) and CYPHEX (right) sites, for the period of measurements (wet season 2014 for T2 at 14 m, July and August 2014 for CYPHEX at 12 m). For T2 a separation between polluted (dotted black line) and clean (thick blue line) air masses was done. Hourly mean diel cycles of temperature and PAR are also shown in red and grey, respectively. Error bars represent the standard deviations.

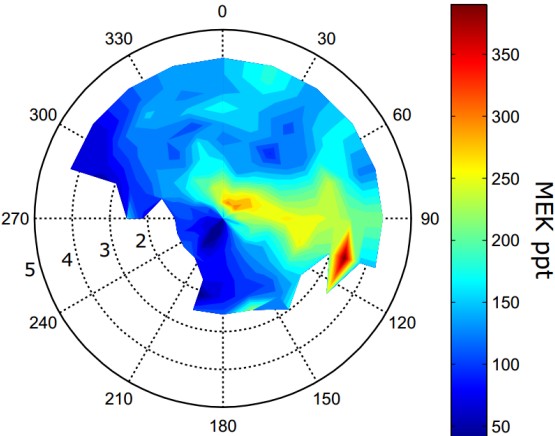

Figure 6: Polar surface plot for average MEK mixing ratios at a given wind direction (angle, 1-5 m s$^{-1}$) and wind speed (radius).





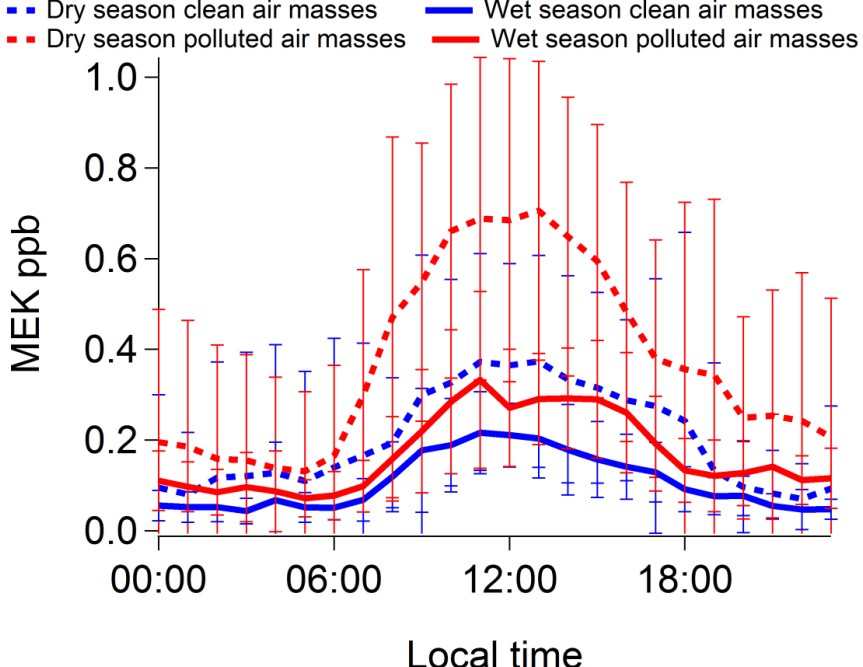

Figure 7: Hourly average concentrations of MEK in ppb for the clean conditions (blue) and the polluted conditions (red) at the T2 site. Dashed lines represent the dry season and thick lines represent the wet season. Error bars represent the standard deviation.

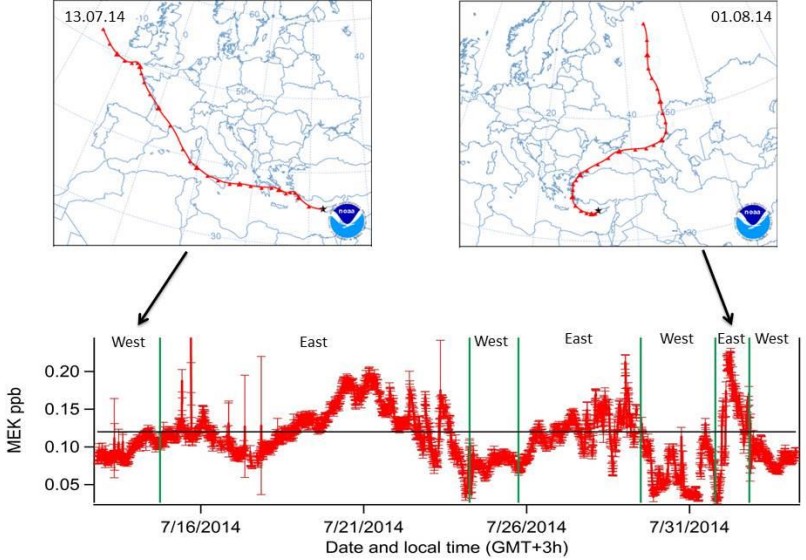

Figure 8: Timeline of MEK mixing ratios divided into periods when the air was coming either from Eastern or Western Europe. The HYSPLIT backward trajectories from 13 July and 1 August. 2014 to show the origin of the air masses. The black line represents the average of the whole campaign.