# Peer review of "Atmospheric mixing ratios of methyl ethyl ketone (2-butanone) in tropical, bo-1"

_Atmospheric Chemistry and Physics, 2016_

## Referee Comment (RC1) · Anonymous Referee #1 · 31 May 2016

The manuscript presents a valuable and solid contribution to the literature on oxygenated VOCs in the atmosphere. I have a number of mostly minor comments that I hope can help further improving the manuscript.

1. Introduction, line 71/72: I think it is unfair to characterize all biomass burning as anthropogenic since a substantial fraction of it is not man-made; naturally occurring fires can be caused by triggers such as lightning.

line 93: I fail to see how MEK can be produced by cis-2-butene or cis-2-pentene. I also did not find that in the cited literature. Please elaborate or drop.

line 121: MEK is of rather low toxicity; it is widely used as a solvent in the US, both in

the chemical industry, and commercially. It is known as "paint-thinner", and available over the counter in home improvement stores such as "Lowes". Aside from its contents in paints and varnishes, it is also used in other household products, such as glues. As the result of such uses, its major fate is evaporation to the atmosphere. Here are some useful sources:

https://www.americanchemistry.com/ProductsTechnology/Ketones/Methyl-Ethyl-Ketone-MEK.html

https://www3.epa.gov/ttn/atw/hlthef/methylet.html

MEK manufacturing and use have been increasing in the last ten years, such that emissions very likely have as well.

2. Results section, line 262f.: While the interpretation that higher MEK mixing ratios near the canopy stem from direct canopy emissions is attractive, a chemical production through fast chemical reactions of emitted terpenes with ozone cannot strictly be excluded. A similar situation was presented by Holzinger et al. (ACP 5, 67-75, 2005). While it would be unusual that MEK, not acetone, is produced from certain terpene structures, the possibility cannot be excluded using the evidence presented. Of course, this origin discussion makes no difference with respect to MEK entering the lower atmosphere. With respect to deposition (and mixing ratios where given), the authors may want to cite Karl et al. (ACP 5, 3015-3031, 2005) and Schade et al. (Biogeochemistry, 106, 337-355, 2011).

line 356: the HYSPLIT citation is incorrect; it should be NOAA not NASA. In fact, the HYSPLIT webpages give concrete instructions on how to cite the model use, which should obviously be followed. The model's use in this case also appears incomplete: several back trajectories for each case should be shown if the preferred ensemble is not used, trajectory heights need to be given, and weather maps should be consulted as a sanity check. This is a major shortcoming in the manuscript.

Interactive
comment

line 361f., section 3.3: This is a poor analysis. It should either be dropped or expanded properly. Simple correlations are not always meaningful, especially in the given setting. Instead, a multi-variate analysis such as a factor analysis (see citation used in line 449) could be performed to evaluate sources, but even in that case, their interpretation may be difficult unless well-defined tracers are available and separated into different factors. In the author's case, the interpretation of acetone, acetaldehyde, and methanol as "biogenic" is questionable since all three compounds, especially acetone and acetaldehyde, have substantial anthropogenic sources. In the case of acetone, its sources are likely very similar to MEK, which makes the exploitation of the MEK to acetone correlation a potentially much more useful variable to explore. The authors barely touch on that in line 375 and 384/385.

3. Discussion section, line 463: "mechanical stress" is undefined. In the particular use here it actually means "cutting", i.e. physically injuring, the plant, which, in nature, is either due to herbivores (common) or infrequent events such as strong storms or heavy snow/frost load (less common). In this respect, I question the formulation of "close agreement" (line 479), since branches or leaves were not cut to enable "the installation of the branch enclosure", or were they? (see also line 546) Anthropogenically cause "wounding" (e.g. deGouw et al. papers) is typically related to crop harvesting.

line 515: lowest MEK abundance at CYPHEX? According to your Figure 5, MEK in "clean" air masses was lower at the T2 site at night. Did you mean averages or medians were lowest at CYPHEX? Also, revise the following section (lines 517f.) after reviewing air mass origins and "correlations".

line 528: What are "night-time anthropogenic activities"? The Guha paper does not list those, but rather links MEK abundances dominantly to soil and agricultural sources.

line 538: It seems the Le Calvé reference is outdated, see above.

5. It would be useful to present a comparison of the MEK data listd in this manuscript and published measurements, such as in Jordan et al and McKinney et al, which the

authors already cite.

---

## Referee Comment (RC2) · Anonymous Referee #2 · 17 Jun 2016

Authors present a study on MEK based on an extensive dataset from six different location. This is a very nice and well-written manuscript on MEK mixing ratios and possible sources with lots of data. It would have been also interesting to see results of other compounds measured with same instruments at the same time. However, this more detailed study on this particular compound is worth publishing also by itself. My recommendation is publishing with minor corrections.

Detailed comments:

It would be interesting to know detection limits and uncertainties for different instruments.

[Figure]

Could you add some comments on seasonal variation of MEK? For example, you mention on lines 275-279 and in discussion that MEK levels were significantly lower at SMEAR-Estonia, but there you were measuring only in October, which is clearly not a high growing season anymore at this site. How was the seasonal variation at the sites where you were measuring for longer period?

Lines 293-297: You state that MEK did not show any covariance with butane and therefore it cannot be related to butane. However, I was wondering, if there is a constant local source or anthropogenic butane emissions are long range transported, then MEK would be produced during the day and lost on surfaces during the night while at the same time butane is not going on surfaces. Then you would not detect any covariance. Maybe you could mention the mixing ratio of butane and it is so low that butane cannot be the main source.

Line 340: change 200 ppt to 0.2 ppb

Lines 365-367: Acetone is regarded as biogenic origin. It has also direct anthropogenic sources and it is produced from the reactions of anthropogenic VOCs also.

Lines 418-420: Something missing from this sentence?

Lines 516-518: Is determination coefficient same as correlation coefficient?

Figure 6. Use ppb as a unit also here.

Table 3: No mean noon mixing ratios are shown and color codes for the sites are missing.
* * *

---

## Author Comment (AC1) · 6 Aug 2016

We acknowledge the work and time provided by this reviewer as all comments truly made the manuscript more consistent and clearer.

1. Comment: It would be interesting to know detection limits and uncertainties for the different instruments. Response: We have added a bracket with LOD and uncertainty next to each site description when available.

2. Comment: Could you add some comments on seasonal variation of MEK? For example, you mention on lines 275-279 and in discussion that MEK levels were significantly lower at SMEAR-Estonia, but there you were measuring only in October, which

is clearly not a high growing season anymore at this site. How was the seasonal variation at the sites where you were measuring for longer period? Response: Most of the sites only have measurements for the periods stated in the manuscript. However, there are some seasonal measurements for ATTO, TT34 and T2. For ATTO there is a paper published commenting on MEK seasonal variation which we have added to the text (Yañez Serrano et al., 2015). For TT34 we were able to have a look at the seasonal change in the mixing ratio raw data to report an approximate seasonal variation. For T2, we already comment about the possible difference in source and sink among seasons when commenting figure 7. Change in manuscript: Line 269: (Holzinger et al., 2005; Karl et al., 2005a). For a seasonal analysis, Yañez-Serrano et al., 2015 reported 0.43 ppb of MEK for the dry season and 0.13 ppb of MEK for the wet season at 38m. Curiously, at 24m, MEK mixing ratios for the wet season were 0.38 ppb, very close to the measured values for this study. Possible differences in temperature and solar radiation among years may be the cause for this variation."; line 274: "In terms of seasonal variation, MEK mixing ratios were observed to be higher during the dry season (∼0.6 ppb) and lower during the wet season (∼0.2 ppb) (data not shown)."; and line 278: "This difference among boreal forests, with growing season ending in October, and broad-leafed tropical (ATTO) and temperate (O3HP) forests could be partly related to the temperature dependence of MEK emissions apparently common among all biogenic sites."

3. Comment: Lines 293-297: You state that MEK did not show any covariance with butane and therefore it cannot be related to butane. However, I was wondering, if there is a constant local source or anthropogenic butane emissions are long range transported, then MEK would be produced during the day and lost on surfaces during the night while at the same time butane is not going on surfaces. Then you would not detect any covariance. Maybe you could mention the mixing ratio of butane and it is so low that butane cannot be the main source. Response: We acknowledge that interesting point raised by the referee and changed the text accordingly as follows below. Change in the manuscript: Line 292: "All samples contained n-butane, which

was of anthropogenic origin with an average mixing ratio of 0.071±0.09 (much lower mixing ratios than MEK), indicating there is no significant source of n-butane nearby."

4. Comment: Line 340: change 200 ppt to 0.2 ppb Response: We have changed it to 0.2 ppb.

5. Comment: Lines 365-367: Acetone is regarded as biogenic origin. It has also direct anthropogenic sources and it is produced from the reactions of anthropogenic VOCs also. Response: We have modified the text to account for extra sources. Change in the manuscript: Line 365 "…site. Using this method, we compare similar compounds to MEK, as this information could indicate some similarities, but this comparison does not necessarily claim links between the various com-pounds. Acetone, acetaldehyde, monoterpenes, isoprene, isoprene oxidation products and methanol are regarded as being of biogenic origin particularly in forested areas (Kesselmeier and Staudt, 1999; Laothawornkitkul et al., 2009). Nevertheless, acetone, acetaldehyde and methanol may have additional sources."

6. Comment: Lines 418-420: Something missing from this sentence? Response: We do not think there is anything missing from this sentence, we are reporting that butanal can fragment on m/z 73, thus influencing the MEK signal, but the manufacturer of the machine has reported to most account for m/z 57.

7. Comment: Lines 516-518: Is determination coefficient same as correlation coefficient? Response: The determination coefficient is the same as the correlation coefficient r2. We just wanted to make sure it was not confused with the pearson coefficient r. For consistency, we will modify all determination coefficient with correlation coefficient (r2).

8. Comment: Figure 6. Use ppb as a unit also here. Response: We have modified the figure to ppb.

9. Comment: Table 3: No mean noon mixing ratios are shown and colour codes for the

sites are missing. Response: This is right; we have removed it from table label.

---

## Author Comment (AC2) · 10 Aug 2016

All authors acknowledge the work and time provided by this reviewer as all comments truly made the manuscript more consistent and clearer.

1. Comment: Introduction, line 71/72: I think it is unfair to characterize all biomass burning as anthropogenic since a substantial fraction of it is not man-made; naturally occurring fires can be caused by triggers such as lightning. Response: This is true and we will change it accordingly. Change in manuscript: Line 71-72: . . ."It is also emitted directly by several anthropogenic sources, including man-made biomass burning (Andreae and Merlet, 2001). . .".

2. Comment: line 93: I fail to see how MEK can be produced by cis-2-butene or cis-2-pentene. I also did not find that in the cited literature. Please elaborate or drop. Response: This is true and we have eliminated cis-2- butane/pentene from the text.

3. Comment: line 121: MEK is of rather low toxicity; it is widely used as a solvent in the US, both in the chemical industry, and commercially. It is known as "paint-thinner", and available over the counter in home improvement stores such as "Lowes". Aside from its contents in paints and varnishes, it is also used in other household products, such as glues. As the result of such uses, its major fate is evaporation to the atmosphere. Here are some useful sources: https://www.americanchemistry.com/ProductsTechnology/Ketones/Methyl-EthylKetone-MEK.html, https://www3.epa.gov/ttn/atw/hlthef/methylet.html MEK manufacturing and use have been increasing in the last ten years, such that emissions very likely have as well. Response: We have taken into consideration this information and have expanded the text with additional information. Change in the manuscript: Line 120: "Most of the urban MEK is released to the atmosphere via evaporation from chemical plants and industrial and household applications, as it is widely used as a solvent (e.g. in glues and as paint thinner). It has a low toxicity, and is not carcinogenic (National Center for Biotechnology, 2015). As its manufacturing has been increasing in the last 10 years, global atmospheric mixing ratios have probably increased as well."

4. Comment: Results section, line 262f.: While the interpretation that higher MEK mixing ratios near the canopy stem from direct canopy emissions is attractive, a chemical production through fast chemical reactions of emitted terpenes with ozone cannot strictly be excluded. A similar situation was presented by Holzinger et al. (ACP 5, 67-75, 2005). While it would be unusual that MEK, not acetone, is produced from certain terpene structures, the possibility cannot be excluded using the evidence presented. Of course, this origin discussion makes no difference with respect to MEK entering the lower atmosphere. With respect to deposition (and mixing ratios where given), the authors may want to cite Karl et al. (ACP 5, 3015-3031, 2005) and Schade et al.

(Biogeochemistry, 106, 337-355, 2011). Response: Even though at ATTO there are low ozone values, the possibility mentioned by the reviewer, although unknown, cannot be totally excluded. Thus, we have decided to incorporate it to the text. In addition, we have included the references mentioned in the text when considering bidirectional VOC exchange and interpretation of temperature to extrapolate MEK as well as acetone. Change in the manuscript: Line 267: "In addition, MEK mixing ratios decreased significantly beneath the canopy towards the forest floor, possibly due to dry deposition or generally smaller vegetation emissions due to less light and temperature. However, a possible production from the ozonolysis of alkanes or bidirectional plant exchange cannot be ruled out."; and line 488: Moreover, a possible production from certain terpenes through ozonolysis cannot be excluded (Holzinger et al., 2005)."

5. Comment: line 356: the HYSPLIT citation is incorrect; it should be NOAA not NASA. In fact, the HYSPLIT webpages give concrete instructions on how to cite the model use, which should obviously be followed. The model's use in this case also appears incomplete: several back trajectories for each case should be shown if the preferred ensemble is not used, trajectory heights need to be given, and weather maps should be consulted as a sanity check. This is a major shortcoming in the manuscript. Response: We acknowledge the mistake in the citation and we have modified it in the text. Furthermore, we now provide the trajectory heights and the ensemble mode used. Weather maps have also been checked. We have also changed Figure 8 to add more information about the settings. Change in manuscript: Line 355: "Furthermore, backward air mass trajectories, as calculated by the HYSPLIT model (NOAA Air Resources Laboratory, USA, Stein et al., 2015) (Figure 8), can be used to delineate times when Cyprus was affected by easterly and westerly flow."; in Line 589: "For CYPHEX: The authors gratefully acknowledge the NOAA Air Resources Laboratory (ARL) for the provision of the HYSPLIT transport and dispersion model and/or READY website (http://www.ready.noaa.gov) used in this publication."; and line 357: "These trajectories were modelled starting at 650 m height with the ensemble mode. The periods (east, west) were chosen on the basis of the FLEXPART model Further information can be

found in Derstroff et al., in preparation.".

6. Comment: line 361f., section 3.3: This is a poor analysis. It should either be dropped or expanded properly. Simple correlations are not always meaningful, especially in the given setting. Instead, a multi-variate analysis such as a factor analysis (see citation used in line 449) could be performed to evaluate sources, but even in that case, their interpretation may be difficult unless well-defined tracers are available and separated into different factors. In the author's case, the interpretation of acetone, acetaldehyde, and methanol as "biogenic" is questionable since all three compounds, especially acetone and acetaldehyde, have substantial anthropogenic sources. In the case of acetone, its sources are likely very similar to MEK, which makes the exploitation of the MEK to acetone correlation a potentially much more useful variable to explore. The authors barely touch on that in line 375 and 384/385. Response: We agree with the reviewer that further statistical analysis would be more enriching, but we also agree with the difficulty of doing so with the variable data sets available. However, we insist on keeping the correlations as the comparison of various compounds to MEK seems interesting to us. However, we have clarified in the text that this comparison does not necessarily claim links between the various compounds. A comment on acetone and MEK similarity was also added. Change in the manuscript: Line 365 "…site. Using this method, we compare similar compounds to MEK, as this information could indicate some similarities, but this comparison does not necessarily claim links between the various compounds. Acetone, acetaldehyde, monoterpenes, isoprene, isoprene oxidation products and methanol are regarded as being of biogenic origin especially in forested areas (Kesselmeier and Staudt, 1999; Laothawornkitkul et al., 2009). Nevertheless, acetone, acetaldehyde and methanol may have additional sources."; and in line 375: "Since sources of both compounds are likely very similar, a high coefficient could indicate similarity in the processes involved in acetone and MEK emission and abundance (Zhou and Mopper, 1993)."

7. Comment: line 463: "mechanical stress" is undefined. In the particular use here

it actually means cutting", i.e. physically injuring, the plant, which, in nature, is either due to herbivores (common) or infrequent events such as strong storms or heavy snow/frost load (less common). In this respect, I question the formulation of "close agreement" (line 479), since branches or leaves were not cut to enable "the installation of the branch enclosure", or were they? (see also line 546) anthropogenically cause "wounding" (e.g. deGouw et al. papers) is typically related to crop harvesting. Response: We agree with the reviewer that such example given in line 479 was not adequate, as the branches were not cut. In this sense, we have decided to remove that part.

8. Comment: line 515: lowest MEK abundance at CYPHEX? According to your Figure 5, MEK in "clean" air masses was lower at the T2 site at night. Did you mean averages or medians were lowest at CYPHEX? Also, revise the following section (lines 517f.) after reviewing air mass origins and "correlations". Response: Yes, we meant lowest averages and have modified in text. We did the same for the correlations. Change in the manuscript: Line 515: "Losses by transport over the sea and chemical decomposition led to the lowest averaged MEK mixing ratios of all compared sites. Correlation coefficients (r2) of MEK with the biogenic tracers were relatively poor for T2 and CYPHEX. However, correlations were also poor for the anthropogenic tracers, although slightly better than at the biogenic sites."

9. Comment: line 528: What are "night-time anthropogenic activities"? The Guha paper does not list those, but rather links MEK abundances dominantly to soil and agricultural sources. Response: We agree with the reviewer that the Guha paper (10.5194/acp-15-12043-2015) links MEK abundances dominantly to soil and agricultural sources, so we have decided to remove such reference.

10. Comment: line 538: It seems the Le Calvé reference is outdated, see above. Response: We agree and we have excluded the comment about human exposure.

11. Comment: It would be useful to present a comparison of the MEK data listed in this

manuscript and published measurements, such as in Jordan et al and McKinney et al, which the authors already cite. Response: We agree with the comment of the reviewer and we will add a table like suggested in the text. Change in the manuscript: Line 533: "Summarizing, Table 4 aims to provide a numerical comparison of MEK mixing ratios reported around the globe. While MEK mixing ratios in our study are relatively constant, MEK has been measured in many different ecosystems ranging from 0.073 ppb to up to 4 ppb. Therefore it is important to account for the variability of this compound."